# Cerebral Sinus Vein Thrombosis and Gender: A Not Entirely Casual Relationship

**DOI:** 10.3390/biomedicines11051280

**Published:** 2023-04-26

**Authors:** Tiziana Ciarambino, Pietro Crispino, Giovanni Minervini, Mauro Giordano

**Affiliations:** 1Internal Medicine Department, Hospital of Marcianise, ASL Caserta, 81024 Caserta, Italy; 2Internal Medicine Department, Hospital of Latina, ASL Latina, 04100 Latina, Italy; 3Emergency Department, Hospital of Lagonegro, AOR San Carlo, 85042 Lagonegro, Italy; 4Advanced Medical and Surgical Sciences Department, University of Campania, L. Vanvitelli, 81100 Naples, Italy

**Keywords:** cerebral venous sinus thrombosis, gender differences, circulation, risk factor

## Abstract

Cerebral sinus venous thrombosis (CSVT) is a relatively rare acute disorder of cerebral circulation, but it can potentially be associated with serious sequelae and a poor prognosis. The neurological manifestations associated with it are often not adequately taken into consideration given the extreme variability and nuances of its clinical presentation and given the need for radiological methods suitable for this type of diagnosis. CSVT is usually more common in women, but so far there are little data available in the literature on sex-specific characteristics regarding this pathology. CSVT is the result of multiple conditions and is therefore to be considered a multifactorial disease where at least one risk factor is present in over 80% of cases. From the literature, we learn that congenital or acquired prothrombotic states are to be considered extremely associated with the occurrence of an acute episode of CSVT and its recurrences. It is, therefore, necessary to fully know the origins and natural history of CSVT, in order to implement the diagnostic and therapeutic pathways of these neurological manifestations. In this report, we summarize the main causes of CSVT considering the possible influence of gender, bearing in mind that most of the causes listed above are pathological conditions closely linked to the female sex.

## 1. Introduction

Cerebral venous sinus thrombosis (CVST) is a rather rare thrombotic disease and its incidence can be considerably underestimated compared to the cases actually present in an emergency setting [1,2]. Acute venous occlusion can lead to increased pressure in small vessels, depending on the extent of thrombosis and the availability of collateral circles [1,2]. This leads to functional alteration of the blood–brain barrier, resulting in vasogenic edema and parenchymal tissue damage, a continuous increase in intracranial pressure results in an increase in capillary pressure which can cause cerebral hemorrhage. The symptomatologic picture of CSVT is extremely variable and is affected by the localization of the veno-occlusive process, the severity of the occlusion to the circulation, and the temporal latency. The most emblematic symptomatic parade is characterized by a headache associated with hemisensory loss, hemiplegia, or hemianopsia affecting one of the two hemispheres. Some more selective cases include the headache associated with visual symptoms such as those cases characterized by papilledema, and paralysis of the oculomotor nerve with a consequent decrease in visus. Other cases may manifest with a prolonged history of headaches preceding an acute event or sometimes the headache can also be associated with the presence of an altered mental state. Sporadically, especially in young patients without atherosclerotic risk factors, the development of acute symptomatology with an epileptic seizure is possible or convulsive crises that are often associated with neuroimaging characterized by infarcts in multiple vascular territories [3,4,5,6,7]. Novel biomarkers indicating the severe brain damage status [8,9] have been recently introduced in clinical practice, although there are difficulties in recognition of venous ischemic processes linked to the complexity of the etiology, and the polyfactorial which includes malignant hematological neoplasms, infectious diseases, pregnancy and the postpartum period, systemic autoimmune diseases, dehydration, intracranial tumors, oral contraceptives, hypercoagulable state, some drugs, trauma, and finally more recently COVID-19 infection and vaccination with adenovirus vaccines. In general, however, it must be said that about 30% of CVST cases still have an unknown etiology [10]. CVST caused by head injury is the rarest condition [10,11]. On the other hand, CVST is very insidious on the basis of underlying or unacknowledged, uncontrolled, or precipitated hyperthyroidism by some acute incidental factors, such as infections, traumas, and surgical interventions. In this case, thyroid storms occur leading to rapid deterioration, involving multiple organs and overall mortality of up to 20–30% [12,13]. Severe trauma is a rare cause of thyroid storms [14,15]. Thyroid storms among trauma patients are difficult to detect because usually, the focus is only on the management of significant post-traumatic injuries and their possible complications, while rarely, symptoms and manifestations such as tachycardia and loss of consciousness are related to an endocrine dysregulation triggered by the trauma. Even a rather frequently observed condition such as thyroid hyperfunction is associated with the genesis of CVST. This observation derives from the fact that at least 20% of patients with CVST have overt or silent manifestations of thyroid disease at diagnosis. This ratio stands at much higher levels if we take into consideration the rate reported by previous observations that have discussed the prevalence of risk factors underlying a CVST manifestation [16,17]. The link between thyroid hyperfunction and CVST, as highlighted for some time in previous studies, can be traced back to a state of aberrant hypercoagulability induced by the excessive increase of circulating thyroid hormones (thyrotoxicosis) [18,19,20,21]. Thyrotoxicosis is a cause of primary and secondary coagulation disorders ranging from an increase in plasma levels of tissue factor, factor VIII, and factor IX, up to an increase in fibrinogen, D-dimer, and activator inhibitor-1 of plasminogen. This set of factors promotes the activation of both the intrinsic and extrinsic pathways of coagulation and especially in the female subject or with other risk factors, it is correlated to a greater risk of acute events attributable to CVST [22,23]. With regard to chronic hematologic malignancies, it has been noted that thrombosis is the leading cause of death and disease in patients with chronic myeloproliferative Philadelphia chromosome-negative disease such as essential thrombocythemia (ET), polycythemia vera (PV), and idiopathic myelofibrosis (IM). Venous thrombosis in atypical sites is frequent and characteristic of these pathologies [24,25]. To date, only a few small studies have evaluated the presence of the JAK2 (V617T) mutation, the Karl gene, and the MPL gene in patients with cerebral venous thrombosis, and the prevalence of these mutations, omitting, however, the presence of gender differences in pathogenesis and therapies both in terms of prevention and treatment of events. In this narrative review, we summarize the main causes of CVST considering the possible influence of gender, bearing in mind that most of the causes listed above are pathological conditions closely linked to the female sex.

## 2. Methods

A bibliographic search was performed on PubMed using the keywords “Cerebral sinus venous thrombosis”, and “Cerebral sinus venous thrombosis and gender difference”. We also searched for data in the literature on the epidemiology, risk factors, pathophysiology, clinical features, diagnosis, and treatment of CSVT in the various pathologies in which gender differences were reported. All English-language full-text articles have been included. Since in some cases CSVT is a rare manifestation, case reports, case series, review articles, and observational studies were analyzed and documented for our review. The literature search we conducted had to be consistent with our goal of performing a comprehensive review of CSVT, with particular attention to forms that may show gender differences in the management of the disease in adults.

## 3. Myeloproliferative Neoplasms and CSVT

Myeloproliferative diseases (MPN) are associated with a greater risk of CSVT and are usually chronic but potentially life-threatening conditions, both for the related complications which include both the tendency to hypo- and hypercoagulable states caused by defective hematopoiesis which also affects the blood coagulation system and clot formation. For the purposes of our discussion, it is important to know that MPNs can be classified as positive or negative for BCR-ABL1, depending on the expression of this gene sequence [1]. In 2005, the somatic mutation in the Janus Kinase 2 (JAK2) gene was described; their replacement of the amino acid valine with the amino acid phenylalanine at codon 617 (V617T) involves increased activity of the JAK2 tyrosine kinase protein and is characteristic of pathological clone in myeloproliferative diseases. The JAK2 mutation (V617T), in fact, is found in 50–60% of patients with ET and IM, and in 95% of those with PV [10,11]. Since 2008, the search for the JAK2 mutation (V617T) has been included as a diagnostic criterion for myeloproliferative diseases in the guidelines of the world health Organization (WHO) and is commonly sought in patients with thrombosis splanchnic. The presence of mutation of the CALR gene is easily found in cases of PMF and ET and is even more evident in youth. It often manifests itself with leukopenia anemia and thrombocytosis [2]. ET is a fairly frequent subgroup of MPN and can be found in a quarter of patients with chronic myelodysplastic manifestations [12]. This clinical condition is characterized by an uncontrolled increase in platelets linked both to a prolongation of the life span of these cellular elements and to an aberrant proliferation of sometimes functionally immature and minimally mature megakaryocytes which introduce cellular elements with altered function into the circulation [13]. In fact, the excessive production of platelets of various levels of maturation and shape, a typical expression of ET, can lead to both thrombotic and hemorrhagic complications since an altered number is also associated with an altered function. The ET is characterized by one long average survival between 18 and 20 years, and it is important to underline that CSVT is more common in females than males and therefore with the concrete possibility of a link between the two conditions mediated by characteristics common to the female gender [14,15]. PV is the most common MPN, accounting for approximately 45% of all MPN cases [12]. It is characterized by an increase in the volume and cell mass of red blood cells and consequently by elevated hemoglobin [16]. In addition, hyperleukocytosis and platelet disease may also be present. These are cellular elements that, despite their excessive numbers, show structural or surface anomalies that affect and cause excessive destruction by the spleen with the consequent finding of splenomegaly [17]. PV is correlated to a very high risk of thrombosis precisely due to an absolute increase in cellular elements circulating in the blood which also tend to agglutinate with each other not only due to excessive concentration but also due to their altered morphology. It is known that the severity and genesis of all these diseases is strongly correlated to the mutation of the tyrosine kinase gene Janus Kinase 2 (JAK2), an important determinant in the estimation of hematopoietic stem cells of the bone marrow. The presence of a mutation in the JAK2 gene is found in almost all patients with PV and in over half of patients with suspected diagnostics of ET and MF [18,19,20,21]. Thus, JAK2 positivity is commonly associated with a more severe course and concurrently with more thrombotic sequelae in patients with MPN.

### 3.1. Risk Factors for Survival and Leukemic or Fibrotic Transformation

A study involving 826 patients showed that median survival was approximately 20 years for TE, 14 years for PV, and 6 years for PMF [22]. The presence of a mutational pattern linked to the JAK2/CALR genes does not negatively affect survival in ET [20]. Conversely, advanced age, leukocytosis, and thrombosis are considered risk factors for survival during ET and PV [19,20]. The 20-year leukemic transformation rate is estimated at <10% for PV and 5% for ET; rates of fibrotic transformation were slightly higher than those of malignant degeneration [19,23]. A population-based study of 327 PV patients in which multivariate analysis identified age > 70 years, severe hyper leukosis, and the presence of preexisting thromboembolic manifestations at diagnosis as risk factors related to a poor prognosis [24]. Furthermore, it was highlighted that the greater the number of risk factors, the lower the survival rate relative to the first 10 years, while the presence of thrombotic manifestations is not among the risk factors of leukemic transformation in PV, including only advanced age, the degree of leukocytosis and the presence of multiple karyotypic abnormalities [21]. Karyotypic abnormalities were the only factor associated with bone marrow fibrosis [25]. In a study dedicated to patients with ET, overall survival was statistically related to the morphology of circulating platelets, advanced age, and the presence of thrombotic episodes, while, unlike PV, the latter together with platelet morphology and extreme thrombocytosis was included among the factors that determined a greater evolution towards an acute form of leukemia [23].

### 3.2. Risk Factors for Thrombosis

As already highlighted, CVST can modify the natural history of patients with myeloproliferative neoplasms (MPN) in a sudden way determining a poor prognosis or correlating the risk of leukemic transformation [26] and the link of both conditions is the JAK2V617F mutation. For the purpose of discussing the role of gender, it emerges that both pathologies taken individually have a predilection for the female sex, but even more significantly, based on the observations of the European Leukemia Net (ELN), it is possible to find a third factor linked to gender namely that the JAK2V617F mutation is more easily detectable in female subjects of young age [26]. The thrombotic risk in lymphoproliferative diseases is the central node of the classifications of this disease. Currently, in PV, two risk categories are considered [17,27,28].

high risk (age > 60 years or history of thrombosis).low risk (absence of both risk factors).

However, for ET, there are four categories:

very low risk (age ≤ 60 years, no history of thrombosis, JAK2/MPL unchanged).Low risk (age ≤ 60 years, no history of thrombosis, JAK2/MPL mutated).intermediate risk (age > 60 years, no history of thrombosis, JAK2/MPL unchanged).high risk (history of thrombosis or age > 60 years with JAK2/MPL mutation).

In a recent study where both arterial and venous thrombosis occurred, predictors of arterial thrombosis included age over 60, history of thrombosis, cardiovascular risk factors including tobacco use, hypertension, diabetes mellitus, leukocytosis, and the presence of JAK2V617F [29]. In contrast, as far as the cases of venous thrombosis are concerned, only the male sex was found to be significantly linked to these events [30]. The relationship between thrombosis and leukocytosis, thrombosis and JAK2V617F6, or pregnancy-associated complications and JAK2V617F has been examined by different groups of investigators with conflicting and inconclusive results [31,32,33].

## 4. Cerebral Vein Thrombosis and COVID-19 Infection

In the case of COVID-19 infection, the greater the severity of the infection, the greater the risk and severity of thromboembolic manifestations, and this could be explained by some physio pathological alterations typical of this virosis, which provide for a direct effect of the virus on endothelial cells, triggering various cascade processes that provide for amplification of acute phase immune processes and inflammation, a decrease in the number of functioning angiotensin-converting enzyme 2 receptors, with consequent blood flow turbulence, and finally the activation of the intrinsic and extrinsic pathways of coagulation with the consequent establishment of a state of hypercoagulability [34,35]. If in the course of COVID-19 acute infection, the mechanisms that determine thromboembolism are quite recognized, and the degree of phenomenon knowledge is different in the post-infectious period since the various studies conducted on the subject show divergent opinions [35,36]. In this regard, a meta-analysis [37] highlighted a risk of venous thromboembolism of just over 10%, which can only be found in patients with an acute form of infection and moreover in the pre-vaccination period. Another case study that included a fair number of studies that compared patients in the post-COVID-19 period with unaffected control groups did not observe statistically significant differences as regards new cases of venous thromboembolism [38]. The increased risk of a first manifestation of thromboembolism was maximal up to three months post-COVID-19 in the case of a deep vein thrombosis and up to six months of acute pulmonary embolism [39]. Regarding the ongoing pulmonary embolism of acute infection with the COVID-19 virus and in the post-infectious period, a significant gender difference was found as regards the number of cases which was significantly higher in men, with a risk that remained high up to the first three months after acute infection and which involved a range of patients aged between 50 and 70 years [38]. In the general population, as regards cerebral venous thrombosis (CVT), cerebral venous sinus thrombosis (CVST) is more common in women than in men, probably due to the participation of specific risk factors in the pathological event of the female gender especially belonging to the younger age groups [39]. The most common risk factors for CSVT concerning young women are pregnancy, use of drugs (oral contraceptives), a pre-existing hereditary thrombophilia state perhaps already associated with previous thromboembolic events, the concomitance of malignant tumors, supervening serious infections, and finally in intra and post-operative period for neurological pathologies susceptible to invasive treatment [39]. Cases of CSVT in patients with COVID-19 have been described more commonly in individuals with severe forms and admitted to intensive care units. From the pathophysiological point of view, the role of any factors is not yet fully clarified—predisposing factors mentioned above or if the cerebrovascular manifestations are caused directly by the virus. It seems clear that pre-existing risk factors of infection act as deterrents of the cerebral embolic process, but there is also much debate about the possible mechanisms that COVID-19 uses to electively cause the involvement of the cerebral venous vessels [40]. At least four hypotheses have been advanced on the mechanism by which the thrombotic process is established:Retrograde cerebral infection starting from the nasal mucosa colonized by the virus and subsequent inflammatory and neuropathic damage of the olfactory nerve [41].Indirect action mediated by the inflammatory hyperactivity triggered by the virulence factors of COVID-19, which would be associated with a real storm of cytokines, dysregulation of the immune system up to the establishment of disseminated vascular coagulation [42].The finding of some autoantibodies circulating during infection would lead to the suspicion of a mechanism of autoimmune genesis which is already the basis of cases of CSVT in the population affected by autoimmune pathologies [43].Direct action of the virus against the vascular endothelium with evidence of histological signs of inflammation and radiological signs of microangiopathy [44,45].

In general, only fourteen cases of CVST have been reported in the literature in patients with COVID-19 which occurred in relatively young patients, with an equal division between the two sexes and in half of the cases, no known risk factors were documentable. As in the general population, most patients complained of headaches, sensory numbness, and confusion associated with focal neurological deficits. Interestingly, more than 60% of CSVT cases had a hemorrhagic transformation and about half of them died due to CVST complications and COVID-19 infection [40]. Ultimately, most probably, COVID-19 acts by means of a multifactorial process, which in conjunction with respiratory and cerebral manipulations is caused by a systemic hypercoagulable state [46]. The main manifestations of CSVTs directly linked to COVID-19 were recorded above all in the pre-vaccination era, and in the most catastrophic phases of the pandemic when it was still looking for the most effective drugs to counteract the manifestations of the disease and was looking for a serum capable of inactivating the virus. Precisely in the latter case, CVT linked to the use of vaccines with adenoviral vectors has been observed [47], among which one of the most devastating clinical manifestations was CVST (Figure 1) [47]. The mechanism of this phenomenon has been clarified and seems to be linked to the presence of a transitory thrombocytopenia vaccine induced by the development of autoantibodies against platelet factor 4 (PF4). The development of these antibodies would lead to a decrease in PF4, to a lower bond with heparin and heparan sulfate produced by the body with the consequent tendency of the platelets to agglutinate each other and to adhere to the endothelial cells. While specifically there were no gender differences in CSVT in patients infected with COVID-19, in this case, female sex and age < 60 years were identified as significant risk factors in subjects immunized with adenoviral vector vaccines [47].

## 5. CSVT Precipitated by Graves’ Disease and Hyperthyroidism

Hyperthyroidism and Graves’ disease, an autoimmune disease involving the formation of stimulating antibodies against the thyrotropin receptor and determining the clinical features of hyperthyroidism, are associated with the risk of CSVT [48,49]. The majority of cases of CSVT related to hyperthyroidism or thyrotoxicosis have been reported in young or middle-aged patients with a predominance of female subjects [49]. In most of these cases, the underlying etiology of hyperthyroidism was Graves’ disease which affects more women than men. In the course of hyperthyroidism, it is possible to note an increase in thromboembolic episodes and this association occurs both in cases of overt thyroid hyperfunction and in cases of subclinical hyperthyroidism [50]. Data available from previous cases showed that patients with hyperthyroidism often exhibit concomitant abnormalities in the coagulation system, such as elevated levels of elevated fibrinogen, factor VIII, factor IX, and von Willebrand factor [51]. This increase in pro-coagulation factors is related to the stimulatory exercise by thyroid hormones [51,52,53,54,55]. The predisposition to thrombotic events would also be linked to platelet activation induced by circulating autoantibodies common to Graves’ disease and other pathologies with autoimmune genesis [49,50]. To confirm this, Erem et al. [56] noted that platelet size and volume correlated with levels of antithyroid peroxidase antibodies and free thyroxine; in addition, platelet volume was correlated with coagulation factor VIII activity. Other evidence has clarified that in hyperthyroidism, both an endothelial dysfunction and an alteration of fibrinolysis are equally present, thus justifying the presence of a state of hypercoagulability in these patients [57]. Taking this evidence into account, it is possible to directly correlate hyperthyroidism with the risk of CSVT. [56]. Knudsen-Baas et al. [58] reported screening for the MTFHR C667T gene mutation in the case of heterozygous CSVT. Mouton et al. [59] reported some cases of CSVT in conjunction with hyperthyroidism showing not only factor VIII impairment but in one case also resistance to activated protein C. Similarly, the same mutation was highlighted in a pediatric case of hyperthyroidism who developed a CSVT in the period following the onset of the thyroid pathology [60]. Considering the Bradford–Hill criteria for all reported cases, it is possible to ascertain a stronger correlation between thyrotoxicosis and CSVT [61,62].

## 6. Trauma and CSVT

The most important pathogenetic mechanism linked to cerebral venous sinus thrombosis (CVST) is linked to intracranial venous blood stasis associated with an activation of the blood coagulation cascade in response to venous endothelial damage [61,62]. Due to the atypical clinical manifestations, CSVT in the course of trauma is often easily overlooked or misdiagnosed. Hyperthyroidism may be one of the predisposing factors for CSVT, but generally in an emergency or emergency medicine setting, knowledge of this correlation is often limited, as well as it being possible that cases of subclinical hyperthyroidism may become manifest precisely in conjunction with the traumatic condition [63]. Frequently, a patient with severe trauma undergoes trauma-related clinical interventions without having adequate attention to hyperthyroidism, which could result in seriously adverse consequences. At the basis of the relationship between uncontrolled hyperthyroidism precipitated by some factors or unacknowledged, and traumas, real thyroid storms have been described which led to rapid deterioration, involving multiple organs and an overall mortality of up to 20–30% [64]. During episodes of thyroid storm, high levels of circulating catecholamines have been described, indicating sympathetic overactivity [65]. On the other hand, catecholamines enhance T4 to T3 conversion in selected tissues, showing a synergistic interaction between thyroid hormones and the sympathetic system [66]. It would appear that thyroid storm enhances sympathetic activity contributing to endothelial damage and then to the development of multiple cerebral thrombotic manifestations.

## 7. CSVT during Subsequent Pregnancy and Puerperium

CSVT is largely a disease of young women and hormonal factors such as taking oral contraceptives or pregnancy are important risk factors in a large percentage of female patients. Hormonal changes during pregnancy and the puerperium led to an increased risk of venous thromboembolism (VTE), including cerebral venous and sinus thrombosis (CSVT) [67,68]. Furthermore, risk assessment of women with a history of CSVT regarding future pregnancies is due to the lack of reliable data on the usefulness of the prophylaxis associated with thrombotic risk. Since pregnancy and the puerperium are prothrombotic risk factors for VTE, often some women with the previous CSVT are discouraged to undertake new pregnancies. However, most studies [67,68] suggest that a history of CSVT is not a decisive factor in precluding a subsequent pregnancy. There is very weak evidence that risk of recurrent VTE for women with an extracerebral history of venous thrombotic events, thromboembolic prophylaxis is required unless there is persistent thrombophilia or if associated with a transient risk factor [67,68]. Conversely, the risk of recurrence increases if a persistent thrombophilia condition is present in these women or if the previous thrombotic episode was ascribed to idiopathic causes. For this reason, women with previous extracerebral or cerebral thrombotic events who are planning to become pregnant should be tested with a comprehensive thrombophilia screening in order to help reduce the individual risk of recurrence during the next pregnancies [67,68]. The decision for prophylactic anticoagulation during pregnancy in women without or without persistent thrombophilia prothrombotic risk factors may be based on the interval between the previous CSVT episode and the next pregnancy [68]. Preter et al. [69] reported an 11.7% recurrence rate of thromboembolic events within the first 12 months of the first episode over a mean follow-up time of over 5 years. From the available data, however, a real risk of recurrence during pregnancy within the first two years after the first event as well as the need for a prophylaxis anticoagulant cannot be estimated. As regards the puerperium period, it has been seen that the number of women treated with anticoagulants has increased considerably, suggesting that the risk of CSVT is also frequent during this phase and not only during pregnancy [70]. Anticoagulant prophylaxis is therefore recommended in women with previous thromboembolic events also in the postpartum period [71,72]. Although these data have been obtained by women with extracerebral thromboembolism, it would appear that similar management should also apply to women with CSVT history.

## 8. Inflammatory Bowel Disease and CSVT

In inflammatory bowel disease (IBD), the incidence of CSVT is quite varied and varies according to reports from 0.5% to 6.7%, and is relatively more common in ulcerative colitis than in Crohn’s disease [73,74]. Approximately half of all patients with ulcerative colitis develop extraintestinal manifestations over the years of the disease, typically arising approximately 15 years after the initial diagnosis [75,76]. Patients with IBD also have a quadrupled risk of thrombosis compared to unaffected people [77]. To date, the cause-effect relationship linking IBD to venous thromboembolism is poorly understood. Remaining in the field of hypotheses, the one that seems more plausible is always represented by the presence and persistence over time of a state of hypercoagulability, which is characterized on the one hand by an increase in pro-coagulant factors such as factor VIII and fibrinogen, higher levels compared to the standard of plasminogen activation inhibitor type 1 (PAI-1), and lipoproteins and a decrease in endogenous anticoagulant factors such as antithrombin III, protein S, and protein C. From a genetic point of view, the polymorphisms of the plasminogen activator inhibitor-1 (PAI-1) gene are a common link between IBD and CSVT [78]. Alongside the state of hypercoagulability, the presence of morpho-structural and functional anomalies of the platelets and endothelial dysfunction due to the inflammatory state have also been reported, which determine flow turbulence and non-natural platelet adhesion [78]. Other alterations have been reported and concern immunological activity due to an increase in circulating antiphospholipid antibodies [79]. It has been observed that thromboembolic events frequently appear during exacerbations or recurrences of ulcerative colitis because the acute inflammatory reaction is the main cause of the hypercoagulable state and therefore of the increased risk of CSVT [80]. The clinical manifestations of CSVT are extremely variable and range from a classic presentation such as a new-onset headache to much more serious nervous manifestations such as convulsions, focal or peripheral neurological deficits, and altered mental status. All these manifestations are linked from the point of pathological examination to an increase in intracranial pressure [81]. Currently, the guidelines on the management of CSVT in patients with IBD are focused on two main objectives such as inflammation control and thrombus dissolution. Some drugs used in the background therapy of IBD such as 5-aminosalicylic acid, azathioprine, 6-mercaptopurine, and infliximab also have the effect of inhibiting platelet activation, while steroids can have an effect in reducing intracerebral edema [82]. The Canadian Association of Gastroenterology also recommends the use of prophylactic anticoagulants in hospitalized patients with severe relapses of IBD or treated outpatient for moderate to severe relapses, especially when there is a history of thromboembolism or an identifiable risk factor [83]. Patients with IBD were younger and more often male compared with those without IBD. Headache was the most common symptom [84].

## 9. Autoimmune Disease-Associated to CSVT

Autoimmune diseases are counted among the diseases most easily predisposing to venous thromboembolism and therefore as a cause of CSVT, due to the presence of circulating immune complexes and the inflammatory state associated with them. Of all the autoimmune diseases, the ones most commonly complicated by CSVT cases are antiphospholipid antibody syndrome, Behçet’s syndrome, systemic lupus erythematosus, and Sjögren’s syndrome [85].

### 9.1. Antiphospholipid Syndrome and CSVT

Antiphospholipid syndrome (APS) is an autoimmune disorder characterized by recurrent arterial and/or venous thrombosis, recurrent spontaneous abortions, recurrent miscarriages, and thrombocytopenia, accompanied by the presence of circulating antiphospholipid antibodies (aPL) [86,87]. This pathology can occur in its primary form when it is characterized only by the presence of anti-phospholipid antibodies or it can be associated with other autoimmune diseases, such as systemic lupus erythematosus (SLE), Sjogren’s syndrome, and some forms of systemic infection [87,88]. Thrombotic events are a major clinical feature of APS, with a prevalence of VTE of 31.7–38.9% [89]. In the course of APS, CSVT is relatively rare and concerns in particular female individuals with a history of single abortion or multiple abortions, use of oral contraceptives, during pregnancy, and in the puerperium phase. Other associations between APS and CSVT have been observed in infectious diseases, malignant tumors, and in the forms of diseases associated with other autoimmune disorders. The key element linking APS to thromboembolism is always to be traced back to a permanent hypercoagulability state [86]. In the case of APS, there is often co-morbidity from infectious and non-infectious causes. A large-scale international study [80] indicated that more than one-third of enrolled patients had a prothrombotic condition which includes antithrombin III, protein C, and protein S deficiency. The presence of circulating aPLs is similarly associated with an elevated risk of CSVT [84]. The aPLs could induce a prothrombotic state through:potentiation of platelet aggregation and adhesiveness through upregulation of thromboxane A223;the amplification of the expression on the surface of the endothelial cells of the expression of the tissue factor capable of activating the extrinsic pathway of the blood coagulation cascade [90];complement activation which in turn interferes with the function of endogenous anticoagulant factors [91].

### 9.2. Behçet Disease and CSVT

Behçet Disease (BD) is a vasculitis of unknown etiology characterized by recurrent mucocutaneous lesions, and ocular lesions. Oral and or genital aphtosis, skin lesions (such as inflammatory papulopustular lesions, erythema nodosum, and skin ulcers) as well as uveitis, retinitis, hypopyon, and iritis are the main manifestations of this disease [92,93]. CSVT constitutes approximately 10% to 20% of Neuro-BD cases and is characterized by thrombosis of the venous sinuses, particularly the superior sagittal sinus, leading to increased intracranial pressure with headache, papilledema, nerve palsy cranial and mental changes [93]. CSVT in BD usually occurs relatively slowly, but the acute onset of seizures and focal neurological symptoms are sometimes observed [92]. One study showed that male subjects had a more frequent history of thrombotic events, malignancies, and Behçet’s disease than females [93].

### 9.3. Systemic Lupus Erythematosus and CSVT

Systemic lupus erythematosus (LE) is a diffuse connective tissue disease that predominantly affects females. CSVT is one of the complications that accompany SLE. Thrombotic events occur in 10–20% of patients with LE, but CSVT is relatively infrequent [85]. CSVT caused by LE usually manifests with aggravated headache, visual field defect or diplopia, impaired consciousness, palpebral and conjunctival chemosis, and epilepsy [85]. The main cause of CSVT in LE cases is the presence of immune-mediated vasculitis. In addition to lupus anticoagulant, antiphospholipid antibodies (aPL) can also be detected in these patients in almost half of the patients. These autoantibodies, when interacting with vascular endothelial cells, inhibit the function of the protein and protein S increasing the risk of thrombosis. Furthermore, concomitant fibrinolytic defects, antithrombin III deficiency, and hyperfibrinogenemia are also pathogenetic factors. Furthermore, a mutation affecting Factor V Leiden has been identified which would seem to be decisive in patients affected by LE with a tendency to thrombosis [94]. Therefore, in patients with LE and CSVT, it is indicated to perform a complete thrombophilia screening including evaluations of protein C and S, lupus anticoagulant, antithrombin III, and the Factor V Leiden mutation [94].

### 9.4. Sjögren’s Syndrome and CSVT

Sjögren’s Syndrome (SS) is an autoimmune disease that is characterized by lymphocytic infiltration of the salivary and lacrimal glands, but there are also cerebral microcirculatory complications such as microcirculatory vasculitis related to white matter lesions and encephalopathy [95]. Several retrospective studies have demonstrated that a certain percentage of SS patients are positive for aPL and lupus anticoagulant. Among patients with aPL, the incidence of stroke and deep vein thrombosis increased significantly. However, the rare episodes of CSVT are not significantly associated with the presence of these antibodies but it is hypothesized that these manifestations may be linked to an immune-mediated nervous impairment [96]. Analyzing all the cases reported in the literature, it is possible to speculate that CSVT during SS completely favors the female sex and therefore it would seem that this is the only variable that, together with the neurological manifestations, can guide in the diagnosis of this complication [96,97,98,99].

## 10. Cerebral Biomarkers

From laboratory tests, in suspected CSVT, it is possible to determine the value of D-dimer although it has a sensitivity of 82–94%, which tends to increase in patients with more extensive and severe disease [100]. Lumbar puncture also provides non-specific outcomes in the field of CSVT research, making it poorly indicated in the diagnostic workup of these cases [101]. It has also been observed that pathological signs at lumbar puncture such as an increase in CSF outlet pressure or the presence of blood have no prognostic value [102]. This method could only play a role in the differential diagnosis of other pathological forms of neurological involvement such as bacterial meningitis or subarachnoid hemorrhage. On the other hand, the contribution of neuroimaging methods is important in the diagnosis of CSVT [101]. Brain CT with intravenous contrast may reveal the “blank delta sign” (not opacified thrombus surrounded by collateral veins of the sinus wall after contrast injection), which although rarely found is pathognomonic with the diagnosis of CSVT, CT or magnetic resonance imaging (MRI) with imaging of the venous phase are techniques diagnostics that have high sensitivity and specificity [102,103]. Magnetic resonance venography shows high sensitivity and specificity for diagnosis and is more accurate in diagnosing all those cases that present with altered mental status and mental confusion that might raise suspicion of involvement of the deep venous system [102,103].

## 11. Conclusions

CSVT is an infrequent form of stroke that often affects younger age groups, particularly women in the reproductive age group. The risk factors are multiple and not always fully known at the time of clinical onset. The clinical presentation is variable and the association with major risk factors contributes to the clinical suspicion. Anticoagulation and recognition of the underlying cause are the mainstays of treatment. Intracranial hemorrhage represents a feared complication although at the moment it is not considered a contraindication to the use of anticoagulants in these patients. Patients with CSVT have an increased risk of recurrence especially within the first 12 months although future studies will be needed to more precisely establish the boundary between the high and low risk of recurrence. Further studies are needed to compare different therapies to improve treatment outcomes of patients with CSVT and lower mortality rates. For this reason, an approach from the point of view of gender medicine can make a valid contribution to optimizing the management of these patients.

## Figures and Tables

**Figure 1 biomedicines-11-01280-f001:**
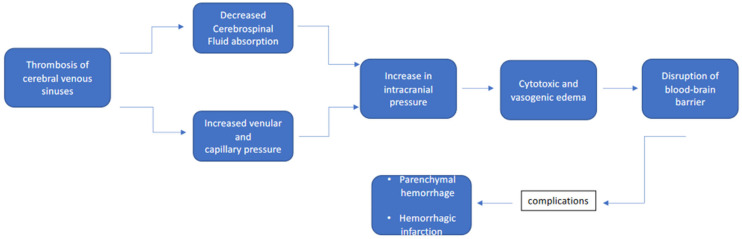
Characteristics and clinical features of CSVT.

## Data Availability

Not applicable.

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
