# Peer review of "Cerebral Sinus Vein Thrombosis and Gender: A Not Entirely Casual Relationship"

_biomedicines, 2023, doi:10.3390/biomedicines11051280_

Round 1

Reviewer 1 Report

The review is based on a broad bibliographic search performed on PubMed using the keywords “Cerebral si-nus venous thrombosis”, and “Cerebral sinus venous thrombosis and gender difference”. The authors give evidence that  an approach from the point of view of gender medicine can make a valid contribution to optimizing the management of patients affected by CSVT thus providing an important overview on CSVT clinical management.

Author Response

Naples, March 13, 2023

Dear Professor Doctor

Biomedicines

Editorial Office

Please, find enclosed the revised version of the manuscript entitled: “Cerebral Sinus Vein Thrombosis and Gender: A Not Entirely Casual Relationship” We thank the Editor and the Reviewers for their comments and we hope that the following changes will now make the manuscript suitable for publication on the Biomedicines. Please see the following list of the underlined changes made in manuscript.

REVIEWER 1

The review is based on a broad bibliographic search performed on PubMed using the keywords “Cerebral sinus venous thrombosis”, and “Cerebral sinus venous thrombosis and gender difference”. The authors give evidence that an approach from the point of view of gender medicine can make a valid contribution to optimizing the management of patients affected by CSVT thus providing an important overview on CSVT clinical management.

Answer: nothing to add

Best regards

Tiziana Cairambino

Reviewer 2 Report

Appraisal

I found the manuscript really interesting and informative. I also believe that the paper can be useful for students and research fellows. I also found very interesting the paragraph about the COVID-19.

The introduction is well written, but I advise to add more details about the symptoms. Moreover, I agree with the authors “Novel biomarkers indicating the severe brain damage status” but a brief  paragraph in the main text about the Cerebral biomarkers (CT , CT perfusion, MRI and DSA) should be needed. In my opinion, this should be interesting and attract more readers. In the methods, despite this is a narrative review, I advise to add the terms and Boolean operators of the search in Pubmed. Indeed, it is quite limiting the use of a single database. Please, justify this choice.

Moreover, given the manuscript is very consistent and informative, Authors need to add a summary or overview table. Similarly, the authors can add a schematic figure that can guide the readers.   

Author Response

Naples, March 13, 2023

Dear Professor Doctor

Biomedicines

Editorial Office

Please, find enclosed the revised version of the manuscript entitled: “Cerebral Sinus Vein Thrombosis and Gender: A Not Entirely Casual Relationship” We thank the Editor and the Reviewers for their comments and we hope that the following changes will now make the manuscript suitable for publication on the Biomedicines. Please see the following list of the underlined changes made in manuscript.

REVIEWER 2

Appraisal

I found the manuscript really interesting and informative. I also believe that the paper can be useful for students and research fellows. I also found very interesting the paragraph about the COVID-19.

The introduction is well written, but I advise to add more details about the symptoms. Moreover, I agree with the authors “Novel biomarkers indicating the severe brain damage status” but a brief  paragraph in the main text about the Cerebral biomarkers (CT , CT perfusion, MRI and DSA) should be needed. In my opinion, this should be interesting and attract more readers. In the methods, despite this is a narrative review, I advise to add the terms and Boolean operators of the search in Pubmed. Indeed, it is quite limiting the use of a single database. Please, justify this choice.

Moreover, given the manuscript is very consistent and informative, the Authors need to add a summary or overview table. Similarly, the authors can add a schematic figure that can guide the readers.  

Answers:

  1. According to the suggestion of the reviewer, more details about symptoms have been added to the text, in the section of the introduction.
  2. According to the suggestion of the reviewer a brief paragraph in the main text about the Cerebral biomarkers (CT , CT perfusion, MRI and DSA) has been added.
  3. According to the suggestion of the reviewer, Boolean operators are words or symbols used as conjunctions to combine or exclude keywords in a search. In reality, some other additional operators were also consulted to perfect the results, but in this case, we obtained the most recent and updated evidence on this specific topic from Pubmed.
  4. According to the suggestion of the reviewer a table and a figure have been added.

Best regards

Tiziana Ciarambino